# Mechanism of Heat Transfer through Porous Media of Inorganic Intumescent Coating in Cone Calorimeter Testing

**DOI:** 10.3390/polym11020221

**Published:** 2019-01-29

**Authors:** Sungwook Kang, J. Yoon Choi, Sengkwan Choi

**Affiliations:** 1Fire Safety Centre, Korea Conformity Laboratories, Cheongju 28115, North Chungcheong Province, Korea; sungwookkang@kcl.re.kr (S.K.); j.yoon.choi@kcl.re.kr (J.Y.C.); 2School of the Built Environment, Ulster University, Newtownabbey BT37 0QB, UK

**Keywords:** intumescent coating, cone calorimeter, heat transfer, porous media, effective thermal conductivity, finite element analysis

## Abstract

This work discusses the heat transfer process through a particular form of porous media: an inorganic-based intumescent coating in full-expansion state. Although the thermal mechanism in porous media has been vigorously studied for polymeric/ceramic/metallic foams, less information is available on its application with intumescent-type polymers. This examination demonstrates the procedure of (1) the optimisation of the coating’s internal multicellular structure for numerical modelling, based on topological analyses; (2) the finite element simulation for the coating-sample tested with cone calorimetry; and (3) the quantitative evaluation of the thermal insulation performance of its porous structure by adopting effective thermal conductivity. The modelling technique was verified using measurable data from the cone calorimeter tests. Consistent agreement between the numerical predictions and experimental measurements was achieved over the whole steel-substrate temperature history, based on the clarified thermal boundaries of the specimen and modelling of the combined conduction-radiation transfer. This numerical approach exhibits the impacts of porosity, pore-size, and external thermal load on the medium’s performance, as well as the individual contributions of the component heat transfer modes to the overall process. The full understanding of this thermal mechanism can contribute to the enhancement and optimisation of the thermal insulation performance of a porous-type refractory polymer.

## 1. Introduction

In the field of structural steelwork, improvements in fire resistance have long been a critical issue, as steel members cease to retain their strength and stiffness at elevated temperatures. This inherent weakness is conventionally addressed through the application of insulation materials to exposed surfaces. Such insulation acts to retard heat transmission to the core structural parts so that the time required for the steel members to reach their critical temperatures can be delayed. Several options are available for passive fire protection such as boards, sprays, blanket systems, and concrete encasement [1]. Among them, intumescent-type coatings have a competitiveness of use given their ability to deliver equivalent fire protection with a reduced thickness, along with a quicker application and better aesthetic finish [1,2]. This type of coating can be categorised into two types: organic-based and inorganic-based (or mineral-based). Although the two groups both have advantages and disadvantages, the traditional organic-type produce toxic fumes upon exposure to heat [3,4,5]. The inorganic-based coating is therefore a viable alternative in compartments where non-toxic products are necessary due to limited ventilation [6].

The aim of this research project was to understand the thermo-physical behaviour of a particular inorganic-type of intumescent coating, tested with a bench-scale cone calorimeter (CC), and to evaluate its thermal insulation performance. This refractory system remains as a thin polymer layer overlying a substrate at ambient temperature, as shown in Figure 1a. Under high temperature conditions, however, it volumetrically expands due to the generation of thermo-chemical decomposition reactions in its internal volume, and in turn becomes a multicellular microstructure, as illustrated in Figure 1b. This distended medium is a rigid foam with a whitish colour (known as the coating residue), rather than black carbonaceous char, due to its main mineral-based ingredients: sodium silicate, kaolinite, aluminium oxide, and titanium dioxide [7]. This foam physically remains until the end of the cone calorimeter tests as it does not reach any of the melting or transforming points of the components (approximately above 1000 °C) during testing. We acknowledge that such geometric formations contribute to the enhancement of the polymer’s ability to retard the heat penetration from its exposed surface area across the distended porous body to the underlying substrate (e.g., the structural steel member). This cognisance notwithstanding, there is a lack of comprehensive interpretation of the heat transfer mechanism through the internal porous structure of this type of fire proofing system.

Heat transfer in porous media has been a popular research topic associated with several engineering applications such as polymeric, ceramic, and metallic foams. This thermal mechanism has been examined by conventionally adopting the principle of the unidirectional transfer of conductive heat flux through porous structures, which is represented by effective thermal conductivity (*k*_eff_). The derivation of this theoretical measure with its verification has thus been a primary objective in relevant studies. This specific topic was of importance in this project and is a prerequisite that must be examined before discussing the overall intumescence of the inorganic polymer. The comprehensive subject of intumescence was thoroughly scrutinised in a companion paper by Kang et al. [8]. As part of the project, we resolved the particular issues associated with understanding the heat transfer mechanism through the intumescent porous medium. To quantitatively analyse its thermal insulation performance, a series of numerical simulations were primarily conducted using finite element analysis (FEA) software (i.e., Abaqus/Standard with user subroutines, Dassault Systemes). To support FEA simulations and their verification, we completed several tests on the characteristics of the polymeric material using bench-scale instruments such as an electric furnace (EF), scanning electron microscopy (SEM), and CC [7]. By using the verified numerical modelling technique, a quantitative analysis was then performed in terms of the individual contributions of the combined heat transfer modes to the overall mechanism.

## 2. Theoretical Background

Heat transfer through a porous medium has been conventionally considered as combined heat flows [9], such as thermal conduction along its solid matrix, thermal radiation across internal pores, and either thermal convection by or conduction through gases filling the pores. With respect to the latter unavoidable selection between convective and conductive heat flux in gas-filled pores, convective heat transfer is suppressed within pores smaller than approximately 10 mm [9,10]. As such, the majority of the pores of the coating-residue studied in this article are expressed on the micron scale, as shown in Figure 1b. In addition to this simple approach, the either/or choice can be completed based on the principle of temperature-fluid motion relations. A previous study [11] stated that the temperature of the heat source in CC (i.e., the spiral coil of the conical heater) remained in the range of approximately 579 to 752 °C at standard irradiance between 25 and 50 kW/m^2^. Under these temperature conditions, the gaseous phase had a Rayleigh number lower than 10^2^, and thus had an average Nusselt number close to unity [12,13,14]. This indicates that the interred generation of buoyancy-induced fluid motions was negligible within the micro-scale pores of the coating-residue. In other words, this heat transfer mode can be regarded as a ‘pure’ conduction through motionless gases. Therefore, in this work, the total heat flux transferring across the fully expanded coating, ${\dot{q}}_{\mathrm{eff}}^{\prime \prime }$, is specified as the heat transfer via solid conduction, ${\dot{q}}_{c\_\mathrm{solid}}^{\prime \prime }$; void conduction, ${\dot{q}}_{c\_\mathrm{void}}^{\prime \prime }$; and void radiation, ${\dot{q}}_{r\_\mathrm{void}}^{\prime \prime }$, as follows:(1)$${\dot{q}}_{\mathrm{eff}}^{\prime \prime }={\dot{q}}_{c\_\mathrm{solid}}^{\prime \prime }+{\dot{q}}_{c\_\mathrm{void}}^{\prime \prime }+{\dot{q}}_{r\_\mathrm{void}}^{\prime \prime }$$

Based on the principle of unidirectional conductive heat transfer, these heat flows are represented by three properties: conductive conductivity of the solid skeleton (*k*_c_solid_); conductive conductivity of gas-filled voids (*k*_c_void_); and radiative conductivity of the voids (*k*_r_void_):(2)$$-k_{\mathrm{cff}}\nabla T=-k_{c\_\mathrm{solid}}\nabla T-k_{c\_\mathrm{void}}\nabla T-k_{r\_\mathrm{void}}\nabla T$$

Russell [15] was among the first to use *k*_r_void_ in defining *k*_eff_ based on the transformation of the algebraic formula from radiative transfer to unidirectional conductive transfer, as follows: (3)$$\begin{array}{ll} {\dot{q}}_{r\_\mathrm{void}}^{\prime \prime } & =\sigma \varepsilon \left\{T^{4}(z+dz)-T^{4}(z)\right\}=\sigma \varepsilon (dz)\left\{\frac{T^{4}(z+dz)-T^{4}(z)}{T(z+dz)-T(z)}\frac{T(z+dz)-T(z)}{(z+dz)-z}\right\}\\ & =\sigma \varepsilon (dz)\frac{\partial T^{4}(z)}{\partial T(z)}\frac{\partial T(z)}{\partial z}=4\sigma \varepsilon (dz)T^{3}(z)\frac{\partial T(z)}{\partial z}=-k_{r\_\mathrm{void}}\frac{\partial T(z)}{\partial z} \end{array}$$

Hereafter, several approximate solutions are proposed to derive correlations of *k*_eff_ with the use of *k*_r_void_ for porous media subjected to different conditions: either the pure conduction [16,17,18,19,20,21] or radiation [22,23,24,25], or both [15,26,27,28,29,30,31,32,33,34,35] were considered in the existing solutions. In terms of the coupling effect between the conduction and radiation modes, this effect is of minor importance for the total heat transfer through porous media [32]. Based on these previous studies, in this work, we assumed the three heat flows (${\dot{q}}_{c\_\mathrm{solid}}^{\prime \prime }$, ${\dot{q}}_{c\_\mathrm{void}}^{\prime \prime }$, and ${\dot{q}}_{r\_\mathrm{void}}^{\prime \prime }$) to be independent, so the corresponding conductivities for the heat fluxes can be expressed as follows: (4)$$k_{\mathrm{eff}}=k_{c\_\mathrm{solid}}+k_{c\_\mathrm{void}}+k_{r\_\mathrm{void}}$$

In this study, the *k*_eff_ measurement of a porous medium represents the ability to transfer (or insulate) heat through the medium’s body, and the components of *k*_eff_ indicate the capabilities to transmit heat in their respective modes. Note that *k*_c_solid_ and *k*_c_void_ do not indicate the true conductivities of the solid-particle and the gas filling pores; rather, they imply the contributions of solid and void conductions to the overall heat transfer in porous media, respectively. Thus, the determinations of *k*_eff_ as well as *k*_c_solid_, *k*_c_void_, and *k*_r_void_, were critical to achieving the objectives of this study. 

In general, the thermal conductivity of matter is a function of temperature and is conventionally measured based on the steady-state guarded hot-plate method (GHM) [36] or the laser flash method [37]. In the case of a present intumescent substance, the physical measurement of its temperature-dependent thermal conductivity is highly problematic, as several thermo-physical properties that considerably affect the measurement of thermal conductivity vary during the course of measurement: the swelling as a reaction to a temperature rise, the length of the conduction transfer path (or thickness) increases, its porosity increases (or density decreases), and its heat capacity per unit envelope-volume decreases. As an alternative method, in this work, a numerical approach is introduced to define the *k*_eff_ versus temperature relation for the particular form of porous medium.

First, a value of *k*_eff_ at the maximum porosity (Φ_max_, in the fully expanded state of the inorganic intumescent polymer) was numerically calculated and verified by experimental data. Second, the initial step was expanded into the determination of a global relationship between *k*_eff_ and porosity (Φ, 0 ≤ Φ ≤ Φ_max_) by adopting the verified numerical technique. Third, the Φ versus temperature relation, which is dependent on the nature of the inorganic intumescent material, was derived based on both the experimental observations and the *k*_eff_(Φ) function determined in this article. The third process is directly related to the simulation of the overall intumescence, which was demonstrated in the companion work by Kang et al. [8]. 

In terms of the technical methodology for the numerical predictions, the combined conduction-radiation transfer through the multicellular body of the fully expanded intumescent coating was simulated using the FEA software: The irregular porous morphology of the coating-residue was numerically modelled as being composed of a structured multicellular formation in the solid-phase (i.e., solid skeleton) and clonal representative elemental cells (REC) in the gas-phase, as illustrated in Figure 2. The solid skeleton represents the dominant routes of the conductive heat transmitted through the solid-phase.The shape and size of the RECs were determined based upon literature reviews of the key geometrical aspects that affect the determination of *k*_eff_ and the results of a topological analysis on the coating-residue’s structural characteristics in terms of the probability distributions of pore size and volume.This numerical simulation also considered the influence of radiative heat absorbing into and emitting from the strut and wall of the pores smaller than the RECs, as illustrated in Figure 2a,b. RECs were therefore regarded as being filled with a mixture of the strut/wall of these smaller pores and air (i.e., a semi-transparent medium).The tortuosity of the solid skeleton was included by considering two outermost cases: inline and staggered configurations, as described in Figure 3. This approach enabled the prediction of the upper and lower limits of *k*_eff_(Φ), which are applicable to the inorganic intumescent system.

To support the FEA, laboratory-scale tests on this material were conducted utilising EF, SEM, and CC. The physical measurements were used to perform the topological analysis on the internal multicellular structure and verify the numerical predictions. Before discussing the primary topic regarding the theoretical methodology and numerical simulations, key information on the experiments are presented in the next section. Tests are outlined only due to the authors’ papers stating their details [7,14].

## 3. Experimental Methods

Although several bench-scale tests were performed, this section introduces two independent experiments that are directly associated with the examination of heat transfer in porous media:(1)Since EF can be maintained at a pre-determined temperature in a steady state, this apparatus was appropriate to identify the relationship of the porosity (or envelope density) versus the temperature, and the distributions of porosity and pore-size along the fully expanded coating’s thickness.(2)The exact quantity of irradiance on the exposed surfaces of a specimen placed in CC was thoroughly scrutinised in a previous study [38]. This knowledge of the sample’s thermal boundaries in CC testing enabled us to concentrate on investigating the interred heat transfer mechanism. Unlike EF, CC simultaneously enables the real-time observation of both volume expansions and temperature increases. These favourable features contributed to the evaluation of the thermal insulation performance of the inorganic intumescent coating as well as to the verification of the numerical predictions.

### 3.1. Electric Furnace Test

#### 3.1.1. Preparation

Plain steel plates, 70 × 70 × 5 mm, were coated with 3-mm-thick inorganic intumescent polymer. The prepared specimens were individually heated from ambient temperature to 800 °C at the rate of 10 °C/min at intervals of 100 °C in an EF. At each level of the pre-planned temperatures, EF’s internal atmosphere was maintained for a certain period of time for the placed sample to reach thermal equilibrium with the EF environment. After full expansion and cooling, the samples were physically sliced (in the x-direction) at intervals of 4 mm using a high-speed cutter. This segmenting was performed to observe the porosity distributions along the coating-thickness (in the z-direction), as illustrated in Figure 4a. The internal structure of each of the bulk (or envelope) volumes was observed by SEM (S-300H SEM Hitachi, Tokyo, Japan), as shown in Figure 4b. 

#### 3.1.2. Topological Analysis

To determine the size of the representative elemental cells (RECs), we completed a topological analysis on the SEM images. Based on the void boundaries highlighted with black lines in Figure 4c, two types of probability distributions were obtained: (1) cell-diameter (*δ*_cell_) and (2) cell-volume (*V*_cell_), as illustrated in Figure 5a. *V*_cell_ indicates the volume occupied by each of the similar-sized groups of pores, if the pore is assumed to be spherical in shape. We found that microscopic pores in a range between approximately 5 μm and 650 μm coexisted. The peak relative frequency densities of *δ*_cell_ (solid lines with round symbols) were observed when the *δ*_cell_ was between approximately 5 μm and 25 μm at both temperatures of 300 and 600 °C. However, the tendency of the relative frequency density distribution of *δ*_cell_ disagreed with that of *V*_cell_ (dotted lines with cross symbols). 

To determine the most representative size of the pores, the relative frequency densities of *V*_cell_ were grouped into seven classes of *δ*_cell_ in 100 μm intervals. Subsequently, relative class frequencies of *V*_cell_ (i.e., *V*_pro_) were derived, as shown in Figure 5b and Table 1. Based on the observations, we concluded that (1) more than 79.5% of the total volume of the void space was accounted for by pores with a diameter of approximately 5 to 400 μm (Classes I, II, III, and IV), and (2) relatively even frequencies were observed in these four classes.

### 3.2. Cone Calorimeter Test

Steel plates, 70 × 70 × 5 mm, were coated with inorganic intumescent polymer with different dry-film thicknesses (DFT) of 2, 3, and 4 mm. These three groups of specimens were individually placed under the conical heater at *z* = 20, 5, and 0 mm, as shown in Figure 6, in order to apply a similar quantity of irradiance to their top surfaces when fully expanded. This initial placement setup was determined through pilot tests. The prepared specimens were consistently exposed to the pre-regulated heat source (i.e., ${\dot{q}}_{\mathrm{irr}}^{\prime \prime }$), which was adjusted to 35, 50, or 65 kW/m^2^ 25 mm underneath the baseplate of the heater in the calibration stage. In the tests, two sets of data were measured: (1) the temperature history of the steel substrate (*T*_s_) using five thermocouples welded on its bottom surface; and (2) the thicknesses of the fully expanded specimens from the recorded video clips using a digital screen ruler, as listed in Table 2. 

## 4. Methodology

### 4.1. Derivation of Effective Thermal Conductivity from Cone Calorimeter Testing

The intumescent fire-proofing system is often tested with CC, as illustrated in Figure 7. In tests, a pre-regulated irradiance (${\dot{q}}_{\mathrm{irr}}^{\prime \prime }$) is consistently imposed on the top surface of a material sample, and part of the absorbed heat is continually emitted by (or lost from) the surface area via radiation and free convection (${\dot{q}}_{\mathrm{rl}}^{\prime \prime }$ and ${\dot{q}}_{\mathrm{cvl}}^{\prime \prime },$ respectively). For a comprehensive interpretation of the *k*_eff_ derivation from CC tests, given the control volume of the porous specimen is regarded as a continuum mixture with an effective thermal conductivity (*k*_eff_), and this continuum is assumed to be composed of two isothermal layers (*V*_top_ and *V*_btm_) with different temperatures of *T*_top_ and *T*_btm_, the heat transfer through the solution domain is governed by three thermal quantities: (1) net heat influx on the top surface of the upper layer (${\dot{q}}_{\mathrm{net}\_\mathrm{in}}^{\prime \prime }$), (2) effective conductive heat flux transferring through the continuum (${\dot{q}}_{\mathrm{eff}}^{\prime \prime }$), and (3) net heat loss on the bottom surface of the lower layer (${\dot{q}}_{\mathrm{net}\_\mathrm{out}}^{\prime \prime }$). The thermal energy stored in each of the layers in differential time intervals (${\dot{Q}}_{\mathrm{st}}$) is then expressed as:(5)$$\begin{array}{c} {\dot{Q}}_{\mathrm{st}\_\mathrm{top}}=\rho_{d}cV_{\mathrm{top}}\frac{dT_{\mathrm{top}}}{dt}=A\left({\dot{q}}_{\mathrm{net}\_\mathrm{in}}^{\prime \prime }-{\dot{q}}_{\mathrm{eff}}^{\prime \prime }\right)\\ {\dot{Q}}_{\mathrm{st}\_\mathrm{btm}}=\rho_{d}cV_{\mathrm{btm}}\frac{dT_{\mathrm{btm}}}{dt}=A\left({\dot{q}}_{\mathrm{eff}}^{\prime \prime }-{\dot{q}}_{\mathrm{net}\_\mathrm{out}}^{\prime \prime }\right) \end{array}$$
where
$${\dot{q}}_{\mathrm{net}\_\mathrm{in}}^{\prime \prime }=\alpha {\dot{q}}_{\mathrm{irr}}^{\prime \prime }-\left({\dot{q}}_{\mathrm{rl}\_\mathrm{top}}^{\prime \prime }+{\dot{q}}_{\mathrm{cvl}\_\mathrm{top}}^{\prime \prime }\right),\;{\dot{q}}_{\mathrm{eff}}^{\prime \prime }=\frac{k_{\mathrm{eff}}}{\ell_{z}}\left(T_{\mathrm{top}}-T_{\mathrm{btm}}\right)$$

From Equation (5), *k*_eff_ can be derived based on the classical principle of the steady-state guarded hot-plate method (GHM) [36], as illustrated in the diagram accompanying Figure 7, as follows: (6)$$k_{\mathrm{eff}}=\frac{{\dot{q}}_{\mathrm{net}\_in}^{\prime \prime }\ell_{z}}{T_{\mathrm{top}\_\mathrm{steady}}-T_{\mathrm{btm}\_\mathrm{steady}}}$$
where
$$k_{\mathrm{eff}}=k_{c\_\mathrm{solid}}+k_{c\_\mathrm{void}}+k_{r\_\mathrm{void}}$$
where *T*_top_steady_ and *T*_btm_steady_ refer to the temperatures of the top and bottom surfaces when reaching a steady state, which correspond to the steady temperatures of the heating and cooling assemblies (*T_h_* and *T_c_*) in the GHM, respectively. 

Equation (6) implies that once the *T*_top_steady_ and *T*_btm_steady_ are obtained, *k*_eff_ can be calculated under the conditions that the external thermal load (${\dot{q}}_{\mathrm{net}\_\mathrm{in}}^{\prime \prime }$) and coating thickness (*ℓ_z_*) are known. In the GHM, the corresponding quantity for ${\dot{q}}_{\mathrm{net}\_\mathrm{in}}^{\prime \prime }$ is identified by recording the average power supplied to the metering section of the heating unit, whereas in this study, it could be thoroughly quantified by theoretically calculating the view factors [38]. The coating thickness is also measurable during cone calorimeter tests, for instance, using a video recorder and a digital ruler [14]. Hence, the two sets of temperature histories of the inorganic intumescent polymer (*T*_top_steady_ and *T*_btm_steady_) were numerically predicted to define *k*_eff_, which in turn represents the given porous medium’s ability to retard the internal heat transfer (i.e., thermal insulation performance).

### 4.2. Scheme of Determination of Effective Thermal Conductivity

#### 4.2.1. Conductive Conductivity of Solid Skeleton

*k*_c_solid_ is dependent on the geometric morphologies of the porous media due to the spatial relationship between the global heat flow direction (from heated to non-heated surfaces) and the orientation of local conduction paths (along solid skeletons); the terminologies are described in Figure 2a. Its determination has conventionally been associated with several idealised cellular structures such as cubical, circular, dodecahedral, and tetrakaidekahedral pores in either open or closed cell forms. This is because the activity of modelling a local exact-morphology of a porous medium neither ensures a representative characteristic nor provides efficiency or cheapness from a theoretical analysis point of view. In recent theoretical models, a representative (or idealised) elemental cell has tended to be determined through topological analyses on two- or three-dimensional visual images, obtained by either X-ray tomography [32] or SEM [39]. Based on the conventional approach to modelling conduction along solid matrices, we developed a simulation scheme to propose a multicellular configuration, optimised for the present inorganic intumescent system. 

There is a noticeable difference between the typical porous media, such as polymeric, ceramic, and metallic foams, and the intumescent coating studied in this paper. The representations of the conventional foams are relatively regular and periodic since their morphologies are deliberately controlled by casting routes in the process of manufacturing. Unlike the intentional formation of these commercial products, the internal structure of the fully expanded coating is constructed in the course of thermo-chemical decomposition reactions, specifically via water vaporisation [14]. The water vapours, liberated from chemical chains, tend to move outward toward the surroundings due to their lower density than that of the residual solid matrix. This rapid migration of water molecules results in a coexistence of both open and closed cells with irregular shapes and sizes throughout the solid matrix. 

As a practical solution method to model this sporadic formation, we only shaped the dominant route of solid conduction in the FEA in this study. Figure 2 describes the scheme of this approach, and illustrates a structured multicellular model superimposed on an SEM image of the fully expanded intumescent coating. The model was composed of a solid skeleton and clonal RECs. With regard to the shape of the RECs, at higher levels of porosity than approximately 0.8, the shapes of the inclusions and cross sections of the struts have minor effects on the thermal conduction along the continuous solid matrices of polymeric media with low conductivity [17,20]. Coquard et al. [20] claimed that numerical models with homogenous dispersions of regular voids enabled the reliable estimation of effective thermal conductivity compared to those with real foams observed using tomographic images. Hence, in this FEA simulation, we modelled the mode of solid conduction as occurring along the structured dominant-conduction-route.

#### 4.2.2. Conductive Conductivity of Gas-Filled Voids

In relation to *k*_c_void_, Qiao et al. [40] claimed that for pores larger than 1.4 μm, the intrinsic thermal conductivity of gases could be adopted disregarding the influence of the reduction in gas conductivity, possibly caused by the correlation between the mean free paths of gas molecules and phonons, called the Knudsen effect. In this work, we therefore simulated the void conduction through the systematic geometry (Figure 2c) by assigning the true conductivity of the air (0.026 W/(mK)) to the gases filling the RECs in FEA modelling. 

#### 4.2.3. Radiative Conductivity of Gas-Filled Voids

At high temperatures, thermal radiation in enclosures, represented by *k*_r_void_, considerably contributes to the overall mechanism of heat transfer in porous media due to the algebraic terms of the fourth power of the absolute temperatures of the surfaces shaping pores [29,32]. The void radiation is influenced by radiative phenomena (absorbing, emitting, and scattering) in association with the optical properties of the struts and walls of pores (extinction, absorption, and scattering coefficients, and scattering phase function) as well as structural characteristics such as porosity, the shape and size of pores, the volume fractions of the pore strut and wall, and their thicknesses. In relation to this heat transfer mode, existing theoretical models can be classified as precise or practical models. The precise models examine the nonlinear distribution of temperature through gas-filled pores caused by the radiant phenomena by adopting the radiative transfer equation (RTE) [21,24,25,31,41]. Practical models prohibit the detailed prediction of temperature gradient across pores but is favourable for identifying the behavioral tendencies of a given porous medium with relatively cheap computational costs [25,27,39].

As the primary purpose of this work was to evaluate the thermal insulation performance of the intumescent polymer’s multicellular structure, the temperature distributions through the internal pores did not need to be specified. Hence, this study conforms to the practical approach to model the void radiation.

The structured solid skeleton with RECs (Figure 2b) efficiently simulates the solid conduction along the dominant heat transfer route. From the radiation viewpoint, however, the radiative phenomena occurring in the pores smaller than the RECs should be considered. In this respect, two cases of RECs in the porous coating-residue were introduced, as shown in Figure 8, to address the issue of the radiative phenomena. The A-type REC was filled only with air, whereas the B-type included smaller pores within its boundary. Unlike the A-type, when radiant heat was transferred through the inner space of the B-type, absorbing, emitting, and scattering occurred on the strut and wall of the smaller pores. These optical events may reduce the amount of the heat transferred through the B-type REC when compared to the A-type. This attenuation effect needs to be checked in the process of theoretical modelling.

In this work, we used the structured solid frame but we assumed its clonal RECs to be filled with an absorbing-emitting medium to satisfy the transparency in the interpretation of the radiative occurrences in the B-type RECs. This semi-transparent medium acts as a mixture of the strut/wall of the smaller pores and the air. This practical approach enabled the simulation of the attenuation effect of cavity radiation based on the principle of radiation in enclosures containing translucent media [41]. Specifically, when a radiant intensity, emitted by an arbitrary *k*th internal surface of a REC (as shown in Figure 9), penetrates the participating medium and arrives at the *n*th internal-surface, its quantity decreases due to absorption by the intervening substance. The absorbed energy increases the mixture’s temperature. The heated medium simultaneously re-emits a radiant intensity to the REC’s internal surface areas. This phenomenon is represented by the geometric-mean transmittance of the participating medium (0 ≤ $\bar{\tau }$ ≤ 1).

The Abaqus/Standard was employed to create the geometry of the structured solid skeleton in the FEA. Although this commercial tool has been optimised for numerical analyses on heat transfer in complex solid geometries via conduction, convection, and radiation, the simulation of the attenuation effect, caused by the addressed semi-transparent medium, is beyond its capability. As an alternative method, we theoretically derived the algebraic matrices of the net radiant heat influx on each of the RECs’ internal surface areas, where the derivation of the matrices is detailed in Section 5.2.1. The calculated heat fluxes were manually assigned to the FEA multicellular model by adopting user-subroutines and the radiant fluxes were continually updated at each time increment. For this FEA, the following assumptions were made:(1)The strut/wall of RECs was opaque, grey, and diffuse.(2)The medium-filling RECs were translucent, isothermal, grey, and diffuse.(3)The two-dimensional FEA model disregarded the volume fraction of the pore-strut/wall and their differences in thickness.(4)The scattering effect in the medium was neglected.

## 5. Numerical Simulations and Results

### 5.1. Modelling Scheme

The combined conduction-radiation through the porous medium, composed of the structured solid skeleton and clonal RECs, was numerically simulated. Two types of FEA models are proposed for different purposes: Figure 10a,b demonstrate the geometry and thermal boundaries of the two types of models:(1)Prototype—This has a single enclosure (*N*_PV_ = 1), as shown in Figure 10a, and was designed to develop a numerical technique capable of simulating the effect of radiation attenuation in microscale pores containing participating media.(2)Multicellular type—The prototype model expands upon multi-enclosures along the z-direction (*N*_PV_ = *n*), as shown in Figure 10b. This was developed to determine the size of the RECs associated with the findings of the topological analysis and to evaluate the thermal insulation performance of the inorganic intumescent system. We completed this assessment by defining a range of effective thermal conductivity (*k*_eff_) that was applicable to the porous coating-residue.

The geometries of the models were determined by three variables: the total thickness of the specimens (*L*), the number of pores in the z-direction (*N*_PV_), and the porosity (Φ), as follows: (7)$$\delta_{\mathrm{cell}}=\delta_{\mathrm{unit}}\sqrt{\Phi }=\left(L/N_{\mathrm{PV}}\right)\sqrt{\Phi }$$
(8)$$\delta_{\mathrm{wall}}=\left(\delta_{\mathrm{unit}}-\delta_{\mathrm{cell}}\right)/2$$

The view factor between the conical heater and the exposed top surface of the models, depending on the top boundary area’s z-coordinate (Figure 6) and the corresponding true radiant heat absorption by this surface area (${\dot{q}}_{\mathrm{net}\_\mathrm{in}}^{\prime \prime }$) are tabulated in Table 3. The principal direction of heat transfer was from the models’ irradiated top surface to the bottom surface. Both side edges of their solid skeletons were assumed to be adiabatic. We simulated conductive heat, transferred along the solid frame and through the air filling enclosures, using the inbuilt procedure of conduction transfer provided by the Abaqus/Standard. The radiant flux (${\dot{q}}_{k}^{\prime \prime }$) on each of the *N* internal-surface areas of the enclosures was calculated and manually assigned using user subroutines. From an FEA viewpoint, both the accuracy of the numerical results and the efficiency of the computations were affected by the number and type of elements. Through a series of sensitivity studies, *N* and element-type were determined as 16 and 8-node biquadratic diffusive heat transfer elements (i.e., DC2D8), respectively. The thermo-physical properties of the materials used in this FEA modelling and the adopted convective heat transfer coefficients are listed in Table 4 [14]. During FEA, the temperature histories of the top and bottom surfaces and the steel substrate (*T*_top_, *T*_btm_, and *T*_s_, respectively) were monitored as these data were critical values for the calculation of *k*_eff_ using Equation (6) as well as verification.

### 5.2. Prototype Model

#### 5.2.1. Derivation of Void Radiation

In order to simulate the radiation transfer through the voids of the inorganic intumescent coating, we formulated a radiation-exchange mechanism in a unit enclosure using the extended net-radiation method [41]. This general form of enclosure was assumed to be composed of *N* discrete internal surface areas and filled with a semi-transparent isothermal medium with *T_g_*, as shown in Figure 9. The energy balance for a solid-phase control volume (*V_k_*), adjacent to the *k*th internal surface area (*A_k_*) of the concave, was derived as: (9)$$\rho_{d}cV_{k}\frac{dT}{dt}=A_{k}\left({\dot{q}}_{\mathrm{ext}}^{\prime \prime }-{\dot{q}}_{k}^{\prime \prime }\right)=A_{k}{\dot{q}}_{\mathrm{ext}}^{\prime \prime }-A_{k}\left({\dot{q}}_{o,\;k}^{\prime \prime }-{\dot{q}}_{i,\;k}^{\prime \prime }\right)$$
(10)$${\dot{q}}_{k}^{\prime \prime }={\dot{q}}_{o,\;k}^{\prime \prime }-{\dot{q}}_{i,\;k}^{\prime \prime }$$
where ${\dot{q}}_{\mathrm{ext}}^{\prime \prime }$ indicates the radiant flux supplied to *V_k_*, ${\dot{q}}_{k}^{\prime \prime }$ represents the net radiative loss from *A_k_*, and ${\dot{q}}_{o,\;k}^{\prime \prime }$ and ${\dot{q}}_{i,\;k}^{\prime \prime }$ indicate the radiant heat fluxes outgoing from and incoming to *A_k_*, respectively. ${\dot{q}}_{o,\;k}^{\prime \prime }$ is composed of the radiant intensities, emitted by and reflected from *A_k_*, as follows:(11)$${\dot{q}}_{o,\;k}^{\prime \prime }=\varepsilon \sigma T_{k}^{4}+\rho_{r}{\dot{q}}_{i,\;k}^{\prime \prime }=\varepsilon \sigma T_{k}^{4}+\left(1-\alpha \right){\dot{q}}_{i,\;k}^{\prime \prime }=\varepsilon \sigma T_{k}^{4}+\left(1-\varepsilon \right){\dot{q}}_{i,\;k}^{\prime \prime }$$
where
$$\varepsilon =\alpha =1-\rho_{r}$$

Based on the principle of cavity radiation in absorbing-emitting media, ${\dot{q}}_{i,\;k}^{\prime \prime }$ is derived using view factor (*F*) and its reciprocity: (12)$$A_{k}{\dot{q}}_{i,\;k}^{\prime \prime }=\sum_{j=1}^{N}A_{j}F_{j-k}\left({\dot{q}}_{o,\;j}^{\prime \prime }\bar{\tau }+e_{b,\;g}\bar{\alpha }\right)=A_{k}\sum_{j=1}^{N}F_{k-j}\left({\dot{q}}_{o,\;j}^{\prime \prime }\bar{\tau }+e_{b,\;g}\bar{\alpha }\right)$$
where
$$F_{j-k}=\left(A_{k}/A_{j}\right)F_{k-j},\;\bar{\tau }+\bar{\alpha }=1,\;e_{b,g}=\sigma T_{g}^{4}$$
where *e_b_*_,*g*_ indicates the energy emitted by the black medium filling enclosures. The subscripts refer to the ordinal numbers of the internal-surface areas (Figure 9), and the dash in the subscript of *F* denotes ‘to’. To eliminate ${\dot{q}}_{i,\;k}^{\prime \prime }$, Equation (11) is substituted into Equation (10), which gives: (13)$${\dot{q}}_{k}^{\prime \prime }=\frac{\varepsilon }{1-\varepsilon }\left(\sigma T_{k}^{4}-{\dot{q}}_{o,\;k}^{\prime \prime }\right)$$

Another form of ${\dot{q}}_{k}^{\prime \prime }$ can be formulated by substituting Equation (12) into Equation (10), as follows:(14)$${\dot{q}}_{k}^{\prime \prime }={\dot{q}}_{o,\;k}^{\prime \prime }-\sum_{j=1}^{N}F_{k-j}\left({\dot{q}}_{o,\;j}^{\prime \prime }\bar{\tau }+e_{b,\;g}\bar{\alpha }\right)$$

Equation (13) is reformulated for ${\dot{q}}_{o,\;k}^{\prime \prime }$ in terms of ${\dot{q}}_{k}^{\prime \prime }$ and *T_k_*, and inserted into Equation (14) in order to quantify the net radiation loss on each of the enclosure’s internal-surface areas (${\dot{q}}_{k}^{\prime \prime }$), as follows:(15)$$\sum_{j=1}^{N}\left(\frac{\gamma_{kj}}{\varepsilon }-F_{k-j}\frac{1-\varepsilon }{\varepsilon }\bar{\tau }\right)\;{\dot{q}}_{j}^{\prime \prime }=\sum_{j=1}^{N}\left[\left(\gamma_{kj}-F_{k-j}\bar{\tau }\right)\sigma T_{j}^{4}-F_{k-j}\bar{\alpha }\sigma T_{g}^{4}\right]$$
where
$$T_{g}^{i+1}=T_{g}^{i}-\frac{dt}{\rho_{d,\;g}c_{g}V_{g}}\left(\sum_{k=1}^{N}A_{k}{\dot{q}}_{k}^{\prime \prime }\right)$$
where *γ_kj_* represents the Kronecker delta, which is equal to 1 at *k* = *j* and to 0 at *k* ≠ *j*; the superscript *i* of *T_g_* refers to the time step.

Subsequently, we built an algebraic matrix using Equation (15) to manually assign the corresponding radiant flux ${\dot{q}}_{k}^{\prime \prime }$ to each of the *N* discrete internal-surface areas of a REC in FEA, as follows: (16)$${\left[A\right]}_{N\times N}{\left\{X\right\}}_{N\times 1}={\left\{M\right\}}_{N\times 1}$$
where
$$\begin{array}{l} \left[A\right]=\left[\begin{array}{cccc} 1/\varepsilon -F_{1-1}\bar{\tau }\left(1-\varepsilon \right)/\varepsilon & -F_{1-2}\bar{\tau }\left(1-\varepsilon \right)/\varepsilon & \cdots & -F_{1-N}\bar{\tau }\left(1-\varepsilon \right)/\varepsilon \\ & 1/\varepsilon -F_{2-2}\bar{\tau }\left(1-\varepsilon \right)/\varepsilon & \cdots & -F_{2-N}\bar{\tau }\left(1-\varepsilon \right)/\varepsilon \\ & & \ddots & -F_{3-N}\bar{\tau }\left(1-\varepsilon \right)/\varepsilon \\ \mathrm{Symmetry} & & & 1/\varepsilon -F_{N-N}\bar{\tau }\left(1-\varepsilon \right)/\varepsilon \end{array}\right]\\ \left\{X\right\}=\left\{\begin{array}{c} {\dot{q}}_{1}^{\prime \prime }\\ {\dot{q}}_{2}^{\prime \prime }\\ \vdots \\ {\dot{q}}_{N}^{\prime \prime } \end{array}\right\},\;\left\{M\right\}=\left[\begin{array}{c} \left(1-F_{1-1}\bar{\tau }\right)\sigma T_{1}^{4}-F_{1-2}\bar{\tau }\sigma T_{2}^{4}-\cdots -F_{1-N}\bar{\tau }\sigma T_{N}^{4}-\bar{\alpha }\sigma T_{g}^{4}\\ -F_{2-1}\bar{\tau }\sigma T_{1}^{4}+\left(1-F_{2-2}\bar{\tau }\right)\sigma T_{2}^{4}-\cdots -F_{2-N}\bar{\tau }\sigma T_{N}^{4}-\bar{\alpha }\sigma T_{g}^{4}\\ \vdots \\ -F_{N-1}\bar{\tau }\sigma T_{1}^{4}-F_{N-2}\bar{\tau }\sigma T_{2}^{4}-\cdots +\left(1-F_{N-N}\bar{\tau }\right)\sigma T_{N}^{4}-\bar{\alpha }\sigma T_{g}^{4} \end{array}\right]\; \end{array}$$

The subscript ordinal numbers of the components in the matrices denote the internal-surface areas, as illustrated in Figure 10a. The inverse matrix of the square matrix [*A*] was obtained using the lower-upper (LU) factorisation. The matrices [*A*] and {*M*} were coupled with the view factor matrix [*F*], which is composed as: (17)$${\left[F\right]}_{N\times N}=\left[\begin{array}{cccc} F_{1-1} & F_{1-2} & \cdots & F_{1-N}\\ F_{2-1} & F_{2-2} & \cdots & F_{2-N}\\ \vdots & \vdots & \ddots & \vdots \\ F_{N-1} & F_{N-2} & \cdots & F_{N-N} \end{array}\right]$$

The view factors between the internal-surface areas were theoretically calculated using the view factor catalogue [41].

#### 5.2.2. Examination of Radiation Attenuation Effect

According to the introduced modelling scheme, the radiation attenuation in enclosures, caused by the absorbing-emitting medium, is represented by the geometric-mean transmittance ($\bar{\tau }$). To identify its effect on the determination of *k*_eff_, this parameter was methodologically altered from 1.0 to 0.2 at intervals of 0.2 under different conditions of porosity (Φ), irradiance (${\dot{q}}_{\mathrm{irr}}^{\prime \prime }$), and pore-size (*δ*_cell_). A $\bar{\tau }$ of 1.0 indicates that no absorption and re-emission exist, and thus enclosures are under a purely transparent condition. Figure 11a shows that $\bar{\tau }$ and *k*_eff_ were in an almost linear relationship. 

Percentage deviation (*D_i_*) was introduced to demonstrate the impact of the $\bar{\tau }$ change on the determination of *k*_eff_, as follows:(18)$$D_{i}\left(\%\right)=\left|k_{\mathrm{eff},\;\bar{\tau }=i}-k_{\mathrm{eff},\;\bar{\tau }=1.0}\right|/k_{\mathrm{eff},\;\bar{\tau }=1.0}\times 100$$

The value of *k*_eff_ at $\bar{\tau }$ = 1.0 was appointed as the reference value. Figure 11b shows a gradual increase in *D_i_* with the reduction in $\bar{\tau }$, obtained in the FEA simulation using the prototype model at *δ*_unit_ = 1 mm. We identified that the more intense the thermal load and the greater the porosity, the larger the *D_i_*. Subsequently, the effect of the $\bar{\tau }$ change was examined according to *δ*_cell_. Figure 11c demonstrates the variations in *D_i_* with the reduction in *δ*_cell_ at different $\bar{\tau }$, under the constant conditions of ${\dot{q}}_{\mathrm{irr}}^{\prime \prime }$ = 50 kW/m^2^ and Φ = 0.90. We observed that the impact of the radiation attenuation on the overall cavity radiation noticeably reduced as the size of the cell decreased from approximately 1 mm to around 10 µm. *D_i_* was less than 6% when *δ*_cell_ was in the range of 5 μm to 400 μm in diameter. Figure 11d shows the absolute values of *k*_eff_ under the identical conditions of ${\dot{q}}_{\mathrm{irr}}^{\prime \prime }$, Φ, *δ*_cell_, and $\bar{\tau }$ to those in Figure 11c. The 6% of the relative measure *D_i_* indicated approximately 0.0082 W/(mK) of *k*_eff_, which was relatively small when compared to the true conductivity of the solid-particle (1.56 W/(mK)). Hence, we concluded that the decrease in the radiation, due to the existence of the intervening mixture of the smaller pores’ strut/wall and the air in pores (B-type REC in Figure 8), can be ignored when the pores are microscopic in size for numerical efficiency and engineering applications.

### 5.3. Multicellular-Type Model

After the sub-study using the prototype model, we conducted a series of primary FEA simulations adopting the multicellular-type model. Two formations of the clonal RECs were introduced: inline and staggered, as shown in Figure 12. From each of the two configurations, two outermost courses of global heat transfer were derived: A-A’ and B-B’ or C-C’ and D-D’, respectively. This approach was designed to consider the tortuosity of the solid conduction routes, and the indirectness of heat flows in void conduction and radiation modes. This allowed us to predict an appropriate range of *k*_eff_ for the given porous structure using Equation (6). 

#### 5.3.1. Determination of Representative Elemental Cell (REC)

In the topological analysis of the SEM images, the applicable range of REC size was narrowed to 5 to 400 μm (Classes I, II, III, and IV), as listed in Table 1. In addition to the observational findings, supplementary numerical simulations were performed to consider the true ability of the four-classed cells to transport heat when determining the most appropriate size of the RECs. Table 5 shows the geometric independent variables input in modelling, and the dependent variables, calculated using Equations (7) and (8). From the simulations, a total of 16 outcomes for *k*_eff_ were produced, as tabulated in Table 6 (four classes multiplied by four heat transfer courses in two types of formations). Next, we weighted the calculated values of *k*_eff_ using the corresponding volume fractions in the four classes (Table 1) to consider their importance by:(19)$$k_{\mathrm{eff}}{}^{*}=\sum_{i=I}^{\mathrm{IV}}\left(k_{\mathrm{eff},\;i}\times V_{\mathrm{pro},\;i}\right)/\sum_{i=I}^{\mathrm{IV}}V_{\mathrm{pro},\;i}$$

The weighted version of *k*_eff_ (i.e., *k*_eff_*) for each of the heat transfer courses and their upper and lower limits are tabulated in Table 7. Notably, the predicted range of *k*_eff_* overlapped that of *k*_eff_ obtained from the multicellular simulation at *δ*_cell_ = 200 μm. The obtained results for *k*_eff_ were in the range of 0.0927 to 0.1117 W/(mK) at Φ = 0.913 and ${\dot{q}}_{\mathrm{irr}}^{\prime \prime }$ = 50 kW/m^2^. Hereafter, 200 μm is used as the representative size of the RECs for the porous coating-residue in subsequent modelling.

#### 5.3.2. Evaluation of Thermal Insulation Performance

The variation in *k*_eff_ according to Φ was examined. Four sets of Φ-*k*_eff_ relations were derived from the courses of A-A’, B-B’, C-C’, and D-D’ in the inline and staggered formations and superimposed on the results obtained using the existing generic models [19], as shown in Figure 13. The conventional models theoretically derived a hybrid of different conductivities of solid and void phases. In contrast, in the multicellular model studied in this paper, *k*_eff_ was derived as a result of numerically simulating the combined mechanism of solid and void conductions and cavity radiation through porous media. We observed that the FEA predictions were positioned between the Maxwell_1 and EMT models in the region of internal porosity. We also found that the true conductivity of the solid-particle at Φ = 0 (i.e., *k*_solid_ = 1.56 W/(mK)) decreased to around 0.0797 W/(mK) at Φ = 0.930. This indicates that the thermal conductivity of the inorganic intumescent polymer decreased by around 5.7% due to the creation of the porous structure in its internal volume. In other words, its thermal insulation performance was noticeably enhanced.

## 6. Discussion

The effective thermal conductivity (*k*_eff_) represents a complex medium’s ability to retard heat penetration. To define the upper and lower bounds of *k*_eff_ for the porous coating-residue, we used Equation (6). (1) The net heat absorbed by its exposed surfaces was accurately quantified by theoretically calculating the view factors. (2) The systematic multicellular model was developed based on the findings identified from the literature review and topological analysis. The model is composed of inline/staggered solid skeleton and clonal RECs of approximately 200 μm in size, which simulates the combined heat transfer via solid and void conductions, and cavity radiation. (3) The proposed model’s top and bottom surface temperatures (*T*_top_ and *T*_btm_ in Figure 10) were numerically predicted.

In this section, we discuss the verification of the developed multicellular modelling and proposed range of *k*_eff_, in association with a complementary multilayer model and the experimental data, which were obtained from the primary cone calorimeter and supplementary electric furnace tests. For validation purposes, the multilayer model is briefly introduced, and details of its development procedure were thoroughly previously discussed [8]. Subsequently, two topics were examined to promote the understanding of the mechanism of heat transfer through porous structures, using the validated multicellular model: (1) the influences of the critical factors for heat transfer in porous media (*δ*_cell_, Φ, and ${\dot{q}}_{\mathrm{irr}}^{\prime \prime }$), and (2) the individual contributions of the component heat transfer modes (solid conduction, void conduction, and void radiation, represented by *k*_c_solid_, *k*_c_void_, and *k*_r_void_, respectively) to the overall heat transfer.

The geometric variables used in these simulations are listed in Table 8.

### 6.1. Verification

Figure 14 illustrates the concept of the model composed of thin layers (i.e., multilayer model), each of which is a discrete and isothermal continuum mixture of solid and void phases. This model was designed based on the observational findings gained from the EF test. In this experiment, we identified a porosity distribution along the depth of the fully expanded coating (Φ_max_) [7]. The corresponding volume fraction (*V_z_*) for each Φ_max_ (or zone) is tabulated in Figure 14. Based on this observation, we assumed that, in a full-expansion state, the Φ_max_ and *V_z_* of any specimen tested with the cone calorimetry reached the listed data observed in the electric furnace test. This assumption was applied in multilayer modelling. The multiplayer model was then coupled with the multicellular model studied in this paper by discriminately assigning the corresponding upper and lower bounds of *k*_eff_ (predicted by the multicellular model as shown in Figure 13) to each of the zones of the multilayer model, as illustrated in Figure 14. As *k*_eff_ at *t*_exp_ is also known as ${\dot{q}}_{{\mathrm{net}}_{\mathrm{in}}}^{\prime \prime }\left(t_{\exp}\right).$
*ℓ_z_*(*t*_exp_), the history of the substrate temperature *T_s_*(*t*), can be predicted by the multilayer model, where *t*_exp_ indicates the instant in time at which the coatings are fully expanded. The predicted profile of *T_s_* was subsequently compared with the measurements obtained from the cone calorimeter tests for verification. 

Figure 15 shows the measurements of *T_s_*(*t*) and *ℓ_z_*(*t*) under different conditions of irradiance (${\dot{q}}_{\mathrm{irr}}^{\prime \prime }$) and dry-film coating thickness (DFT) as obtained from the CC tests. An important finding was that the full-expansion time *t*_exp_ was similar to the moment in time at which *T_s_* re-increased after undergoing a short plateau. This indicates that, hereafter, *T_s_* development was governed by the mechanism of heat transfer through the fully expanded coating (rigid porous residue). Therefore, the following history of *T_s_* beyond *t*_exp_ can be used for the validation of the multicellular modelling. Figure 16 demonstrates both the *T_s_* histories, physically measured from the cone calorimeter test and numerically predicted by the multilayer model coupled with the multicellular model. We found that the FEA model accurately and consistently predicted *T_s_*(*t*) under the different conditions from the transient to steady states. Notably, the graphs in the shaded regions (<*t*_exp_) are not directly related to this verification objective; these are associated with the process of the polymer’s mass and volume changes depending on temperature, which were thoroughly examined by Kang et al. [8].

### 6.2. Effect of Cell Size

Figure 17 demonstrates the influence of *δ*_cell_ on the thermal insulation performance of porous structures. Its effect was quantitatively expressed by the ratio of *k*_eff_ to the true thermal conductivity of the solid-particle (*k*_solid_). The contributions of *k*_c_solid_, *k*_c_void_, and *k*_r_void_ to this ratio are also specified in each of the bars of the chart. The red and black colours denote the upper and lower bounds of the FEA predictions, respectively, derived from the four heat transfer courses (A-A’, B-B’, C-C’, and D-D’). At *N*_PV_ = 0, heat was transferred via pure solid conduction. As *N*_PV_ increased at a constant *L*, *δ*_cell_ reduced to around 60 μm and the ratio noticeably decreased to the range of approximately 0.075 to 0.065. 

When the size of the cell was in millimetres (≳1000 μm), a large portion of the *k*_eff_/*k*_solid_ ratio was accounted for by *k*_r_void_, whereas *k*_c_void_ made a minor contribution. The reasons for these observations are that radiation is on the basis of the fourth power of the absolute temperature, and the true thermal conductivity of the air is much smaller than *k*_solid_. However, the contribution of the cavity radiation was dramatically reduced with decreasing *δ*_cell_. Around the determined *δ*_cell_ for the inorganic intumescent coating (200 μm), solid conduction became the major mode of heat transfer through the porous structure. We interpreted that with the decrease in *δ*_cell_, the temperature difference between the hot and cold internal surface areas of RECs (i.e., Δ*T*) considerably narrowed. This trend caused a reduction in the net radiant flux (${\dot{q}}_{k}^{\prime \prime }$) absorbed by each of the internal-surface areas, when being transported from one surface to another and vice versa within the RECs (Figure 10a). Figure 18 demonstrates the quantities of ${\dot{q}}_{k}^{\prime \prime }$ and Δ*T* on two internal surface areas with the highest and lowest temperatures (*N* = 11 and 2, respectively) of each of the two outermost RECs (*N*_PV_^th^ and first RECs in Figure 10b). We found that ${\dot{q}}_{k}^{\prime \prime }$ and Δ*T* were less than approximately 0.4 kW/m^2^ and 4 °C at *δ*_cell_ = 200 μm.

### 6.3. Effect of Porosity

Figure 19 shows the variations in the ratio of *k*_eff_ to *k*_solid_ and the individual contributions of *k*_c_solid_, *k*_c_void_, and *k*_r_void_ to the ratio, according to Φ. The greater the porosity, the lower the *k*_eff_/*k*_solid_ ratio, and the more enhanced the thermal insulation performance of the porous medium. This tendency was induced by the considerable reduction of the solid-conduction contribution, which was associated with the decrease in the volume fraction of the solid-phase with the increase in Φ. With this decrease in the volume fraction of the solid-phase, an increase in the volume fraction of the void-space was produced. Little change, however, was observed in terms of the contribution of *k*_c_void_. This was closely related to the low intrinsic conductivity of the air compared to *k*_solid_. With respect to *k*_r_void_, since *δ*_cell_ did not change in this case study, little variation was found in the contribution of *k*_r_void_, which was strongly dependent on pore size, as shown in Figure 17.

### 6.4. Effect of External Thermal Load

Figure 20 illustrates the impact of ${\dot{q}}_{\mathrm{irr}}^{\prime \prime }$ on the determination of the *k*_eff_/*k*_solid_ ratio as well as the individual contributions of the three heat transfer modes. Overall, the more intense the external thermal load, the greater the ratio. This indicates that the thermal insulation performance of the cellular structure can vary according to given heating conditions. Cavity radiation was more dependent on the change in the thermal load than the other heat transfer modes. This feature is also based on the fundamental principle of thermal radiation, being defined with the fourth power of the absolute temperature.

## 7. Conclusions

The main purpose of this work was to define the range of effective thermal conductivity (*k*_eff_) applicable to the particular form of porous medium (inorganic intumescent coating) to quantitatively assess its thermal insulation performance. The measurement *k*_eff_ was numerically derived from finite element analysis (FEA). In the course of its derivation, we simulated the combined conduction-radiation transfer through the hybrid of solid and void phases using the Abaqus/Standard with user subroutines. 

In both the quantitative evaluation of *k*_eff_ and the quantitative analysis of the individual contributions of the solid-conduction, void-conduction, and void-radiation modes, the controlling factors were porosity, pore-size, and external thermal load. We identified that amongst the factors, the alteration of porosity led to the most dramatic change in the ratio of *k*_eff_ to the true conductivity of the solid-particle (*k*_solid_). At the maximum porosity of 0.930, the ratio dropped by approximately 5.1%, which quantitatively demonstrated the enhancement in the coating’s thermal insulation performance. This occurred because an increase in porosity indicates the growth in the volume fraction of the void spaces filled with air, the true thermal conductivity of which is much smaller than *k*_solid_. The decrease in pore-size to the micro-level and the lower thermal load caused a reduction in the radiant flux transferring across the pores due to the decrease in the temperature difference between the relatively hot and cold internal surface areas of the micro-scale enclosures.

Based on the findings, we conclude that an increase in porosity, a decrease in pore size, a lower level of thermal load, and a decrease in *k*_solid_ can contribute to enhancing the thermal insulation performance of polymeric porous media; and the impact of translucent media filling pores was relatively weak on the condition of the microscopic pores. So, whether the assumption of neglecting the radiation attenuation effect is acceptable for macroscopic pores needs to be carefully considered.

The numerical predictions were verified by the experimental data. Consistent agreement between the FEA predictions and cone calorimeter test measurements was achieved in terms of the temperature history of the underlying steel substrate. This was the only reliable output of the temperature measurements in the cone calorimeter testing. This agreement could be attained based upon (1) the clarified thermal boundaries of the coating-residue, tested with cone calorimetry; (2) the design of the FEA model’s geometry, depending on the topological analysis; (3) the simulation of the combined conduction-radiation transfer; and (4) the attempt to define the applicable range of *k*_eff_, instead of a single value by considering the irregularity and tortuosity of the coating-residue’s internal structure. Although several attempts have been made to propose the *k*_eff_ of an intumescent coating’s porous structures in the field of fire safety engineering, comprehensive interpretation is lacking for its derivation based on the theoretical principle of heat transfer in porous media to its verification with practical engineering applications (e.g., bench-scale experiments). This numerical approach was able to demonstrate the process, and therefore contributes to the full understanding of heat transfer in the porous intumescent system. The quantitative evaluation of the coating’s performance and the quantitative analysis on the contributions of the heat transfer modes provide insights into the development of porous-type refractory systems and the optimisation of their fire resistance in various engineering fields.

## Figures and Tables

**Figure 1 polymers-11-00221-f001:**
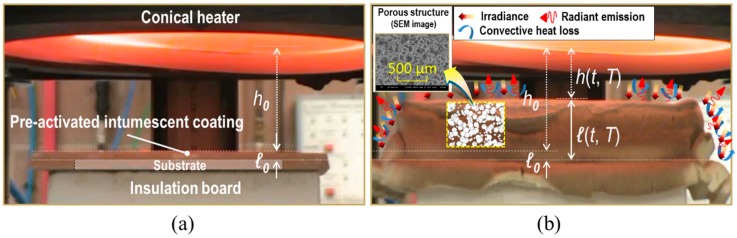
Thermo-physical behaviour of an inorganic-based intumescent coating tested with cone calorimetry: (**a**) in the initial state and (**b**) in the fully-expanded state, where *ℓ*, *h*, and SEM refer to the thickness of the coating, the distance between the heater and the coating’s exposed surface, and scanning electron microscopy while the subscript 0 indicates the initial state.

**Figure 2 polymers-11-00221-f002:**
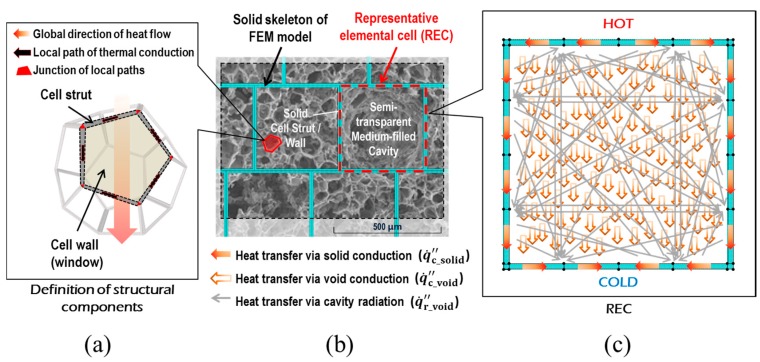
Modelling scheme for heat transfer in porous media: (**a**) definition of the structural components of a cell, (**b**) conceptual solid skeleton superimposed on an SEM image, and (**c**) heat flows in a representative elemental cell (REC).

**Figure 3 polymers-11-00221-f003:**
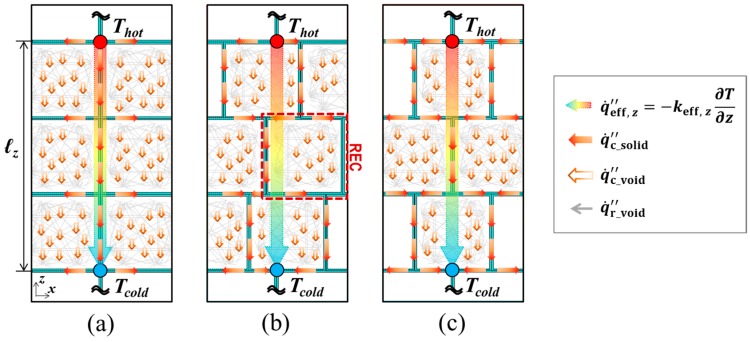
Tortuosity of the solid skeleton: (**a**) outermost inline, (**b**) in-between, and (**c**) outermost staggered formations.

**Figure 4 polymers-11-00221-f004:**
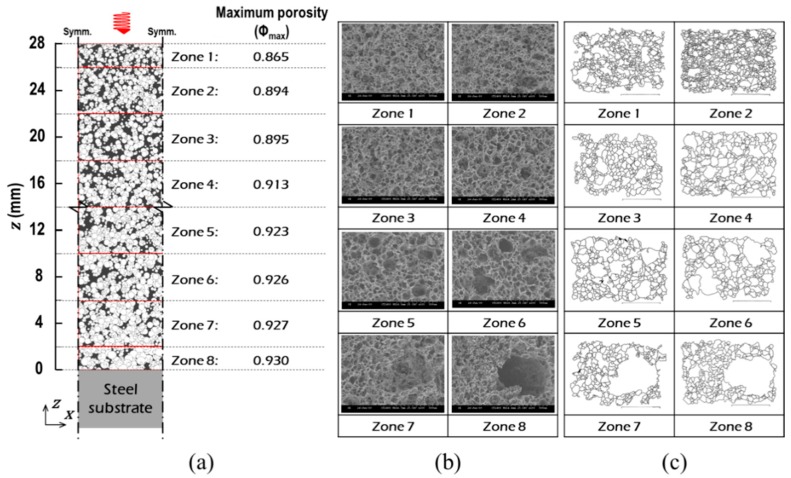
Data obtained from the electric furnace tests: (**a**) schematics of the porosity distribution along the coating thickness, (**b**) the SEM images, and (**c**) the corresponding void boundaries.

**Figure 5 polymers-11-00221-f005:**
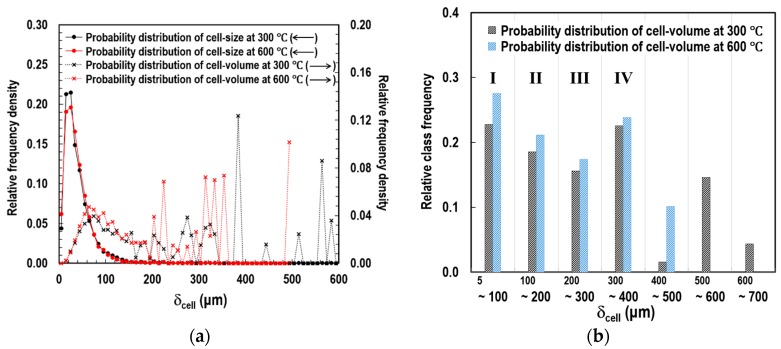
Probability distributions of cell size and volume: (**a**) relative frequency densities of *δ*_cell_ and *V*_cell_, and (**b**) relative class frequency of *V*_cell_.

**Figure 6 polymers-11-00221-f006:**
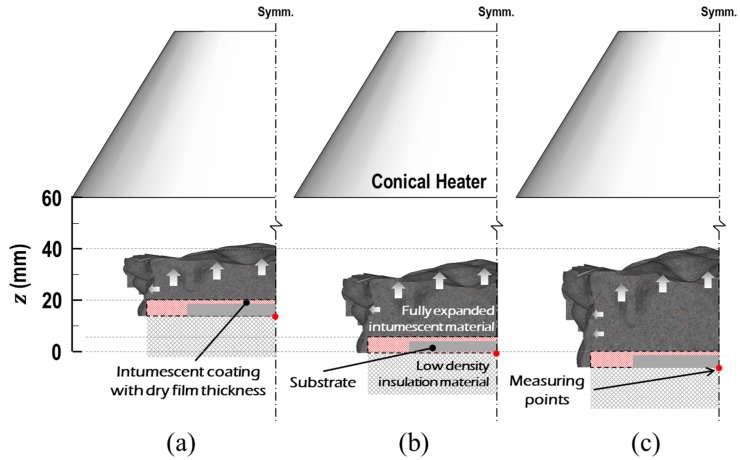
Initial setup of the specimen placement with a dry-film thickness of (**a**) 2 mm, (**b**) 3 mm, and (**c**) 4 mm.

**Figure 7 polymers-11-00221-f007:**
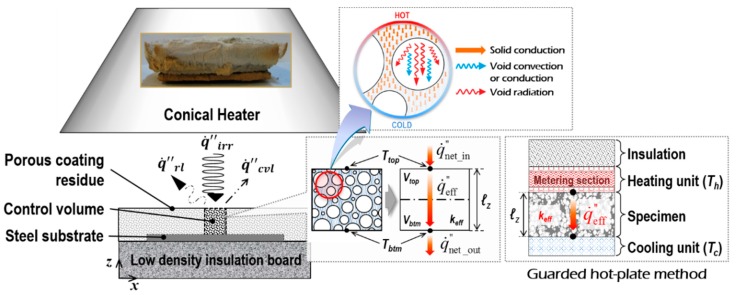
Heat transfer mechanism through the porous specimen in cone calorimeter testing.

**Figure 8 polymers-11-00221-f008:**
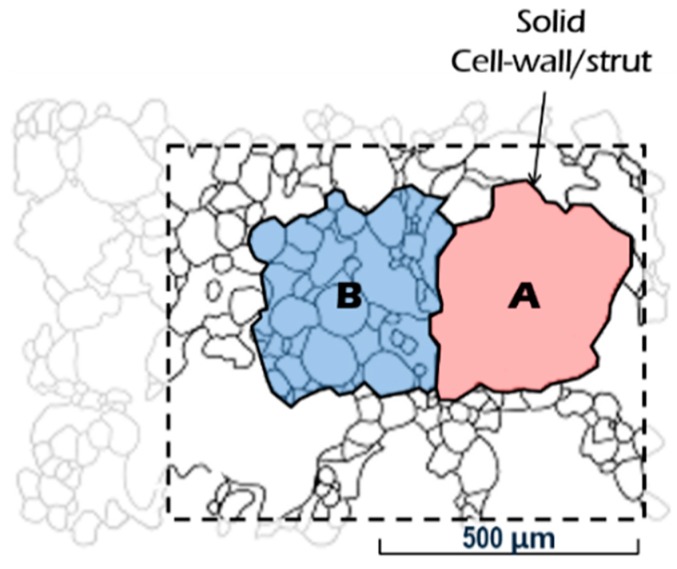
Modelling scheme for the radiation in pores.

**Figure 9 polymers-11-00221-f009:**
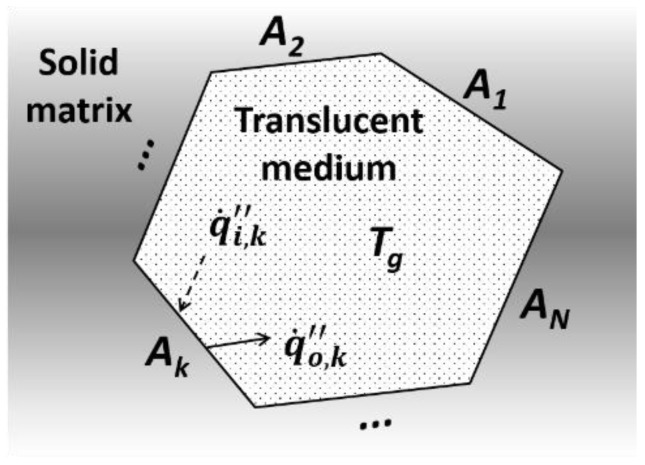
Unit enclosure, which is composed of *N* discrete elemental surfaces and filled with translucent media.

**Figure 10 polymers-11-00221-f010:**
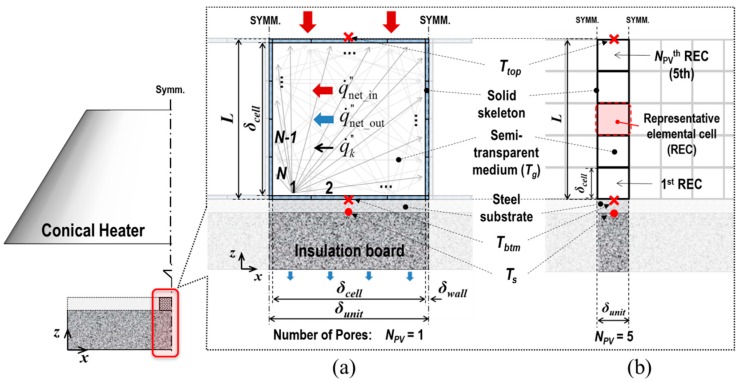
Scheme of the FEA modelling: (**a**) prototype (*N*_PV_ = 1) and (**b**) multicellular-type (e.g., *N*_PV_ = 5 in inline formation).

**Figure 11 polymers-11-00221-f011:**
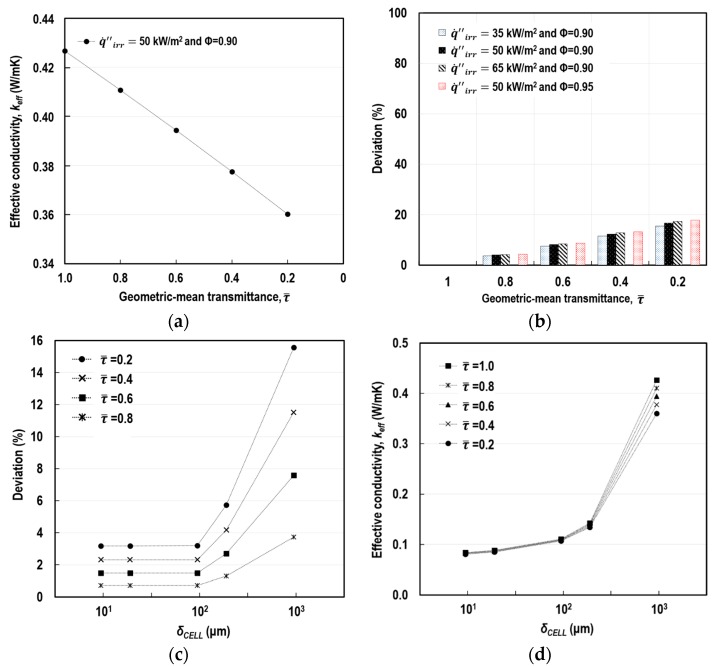
Effect of semi-transparent media on cavity radiation: (**a**) relation between $\bar{\tau }$ and *k*_eff_ at ${\dot{q}}_{\mathrm{irr}}^{\prime \prime }$ = 50 kW/m^2^ and Φ = 0.90; (**b**) deviations from *k*_eff_ at $\bar{\tau }$ = 1.0 with changes of $\bar{\tau }$, ${\dot{q}}_{\mathrm{irr}}^{\prime \prime }$, and Φ; (**c**) deviations from *k*_eff_ at $\bar{\tau }$ = 1.0 with changes of *δ*_cell_ and $\bar{\tau }$, at ${\dot{q}}_{\mathrm{irr}}^{\prime \prime }$ = 50 kW/m^2^ and Φ = 0.90; and (**d**) *k*_eff_ at ${\dot{q}}_{\mathrm{irr}}^{\prime \prime }$ = 50 kW/m^2^ and Φ = 0.90 with changes of *δ*_cell_ and $\bar{\tau }$.

**Figure 12 polymers-11-00221-f012:**
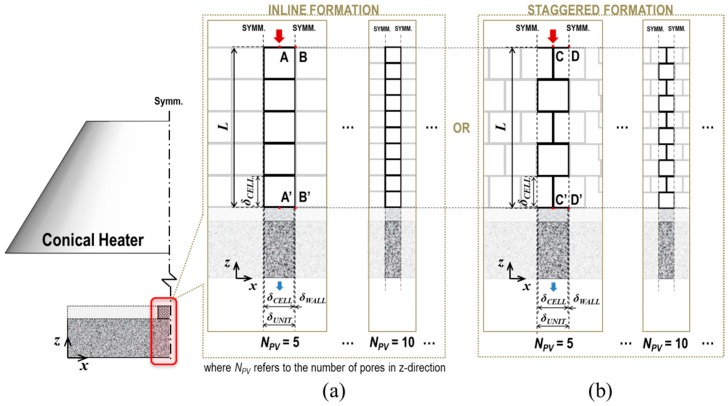
Scheme of FEA multicellular modelling: (**a**) inline and (**b**) staggered formations.

**Figure 13 polymers-11-00221-f013:**
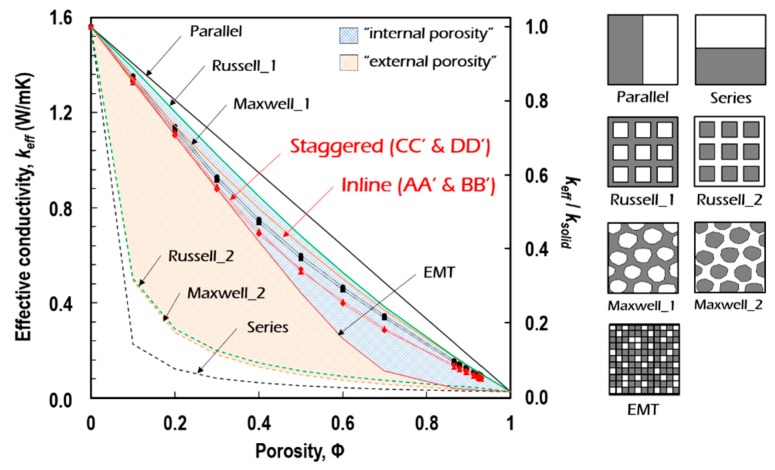
Φ-*k*_eff_ relationships obtained from FEA multicellular simulations with existing generic models.

**Figure 14 polymers-11-00221-f014:**
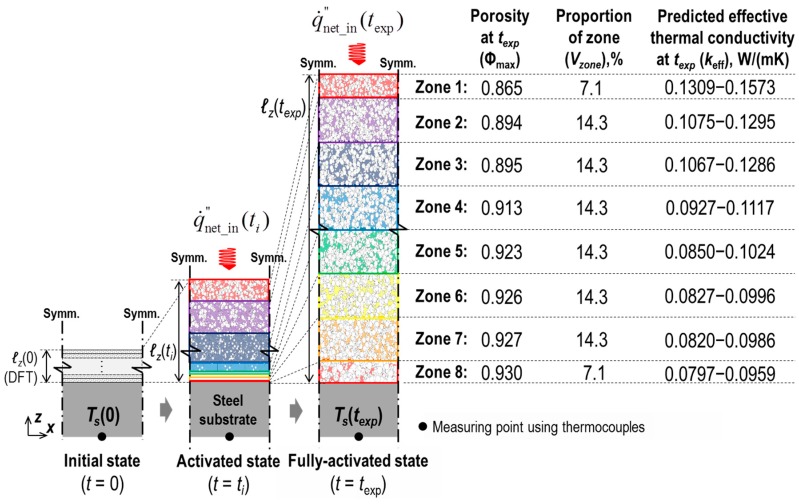
Schematic of concept of the FEA multilayer model for intumescence over time in association with the electric furnace test.

**Figure 15 polymers-11-00221-f015:**
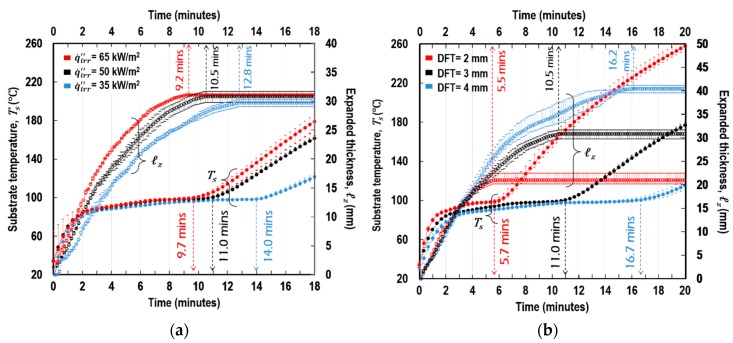
Experimental measurements of substrate-temperature (*T_s_*) and thickness expansion (*ℓ_z_*) over time: (**a**) at DFT = 3 mm and (**b**) at ${\dot{q}}_{\mathrm{irr}}^{\prime \prime }$ = 50 kW/m^2^.

**Figure 16 polymers-11-00221-f016:**
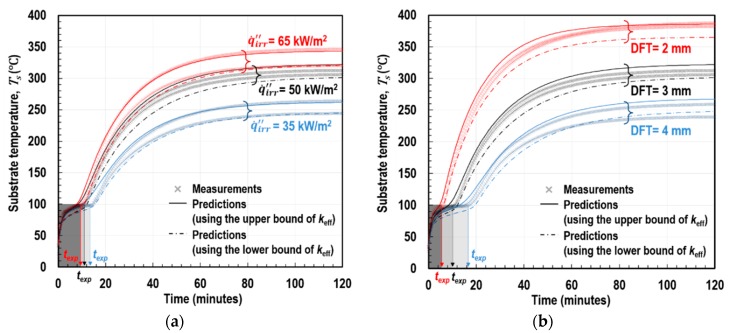
Experimental measurements and numerical predictions of the upper and lower limits of substrate-temperature (*T_s_*) and thickness expansion (*ℓ_z_*) over time: (**a**) at DFT = 3 mm and (**b**) at ${\dot{q}}_{\mathrm{irr}}^{\prime \prime }$ = 50 kW/m^2^.

**Figure 17 polymers-11-00221-f017:**
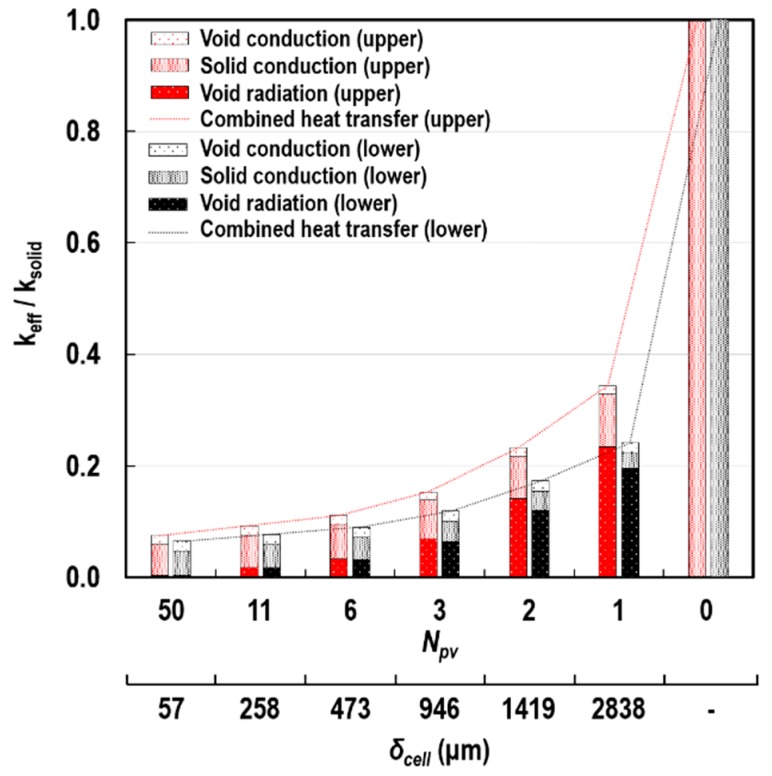
Variations in the *k*_eff_/*k*_solid_ ratio and the individual contributions of the heat transfer modes according to *δ*_cell_ at *L* = 3 mm, Φ = 0.895, and ${\dot{q}}_{\mathrm{irr}}^{\prime \prime }$ = 50 kW/m^2^.

**Figure 18 polymers-11-00221-f018:**
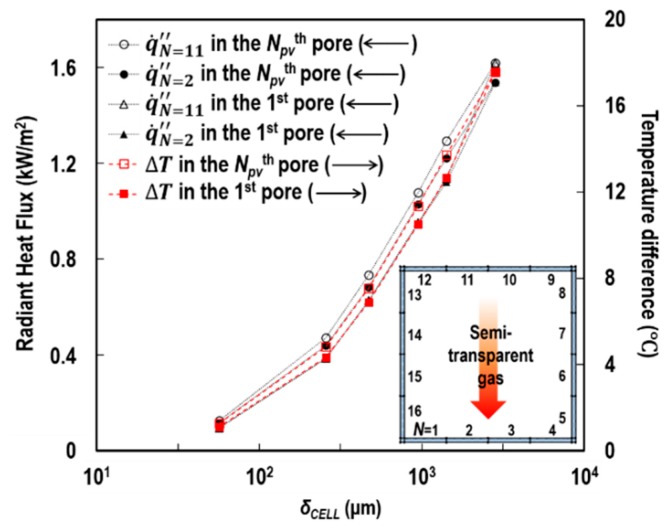
Variations in ${\dot{q}}_{k}^{\prime \prime }$ and Δ*T* according to *δ*_cell_, at *L* = 3 mm, Φ = 0.895, and ${\dot{q}}_{\mathrm{irr}}^{\prime \prime }$ = 50 kW/m^2^.

**Figure 19 polymers-11-00221-f019:**
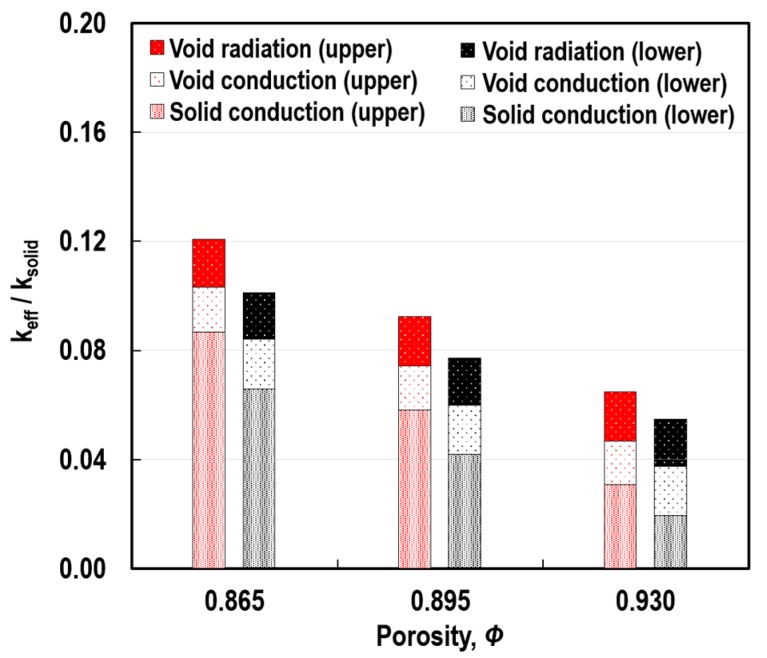
Variations in the *k*_eff_/*k*_solid_ ratio and the individual contributions of the heat transfer modes according to Φ at *L* = 3 mm, *δ*_cell_ = 258 μm, and ${\dot{q}}_{irr}^{\prime \prime }$ = 50 kW/m^2^.

**Figure 20 polymers-11-00221-f020:**
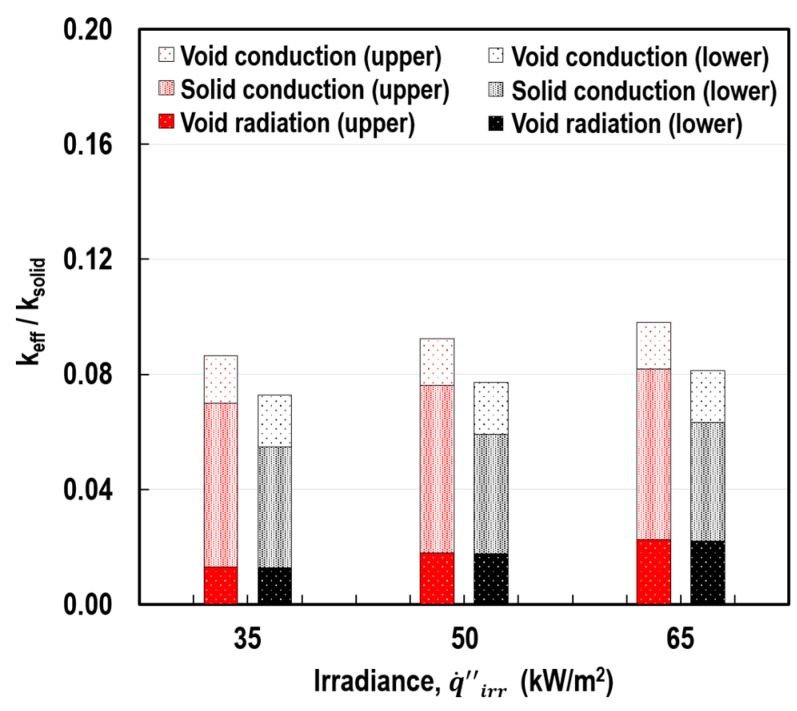
Variations in the *k*_eff_/*k*_solid_ ratio and the individual contributions of the heat transfer modes according to ${\dot{q}}_{\mathrm{irr}}^{\prime \prime }$ at *L* = 3 mm, *δ*_cell_ = 258 μm, and Φ = 0.895.

**Table 1 polymers-11-00221-t001:** Percentage probability distributions of cell volume.

Temperature (°C)	Probability Distributions, *V*_pro_ (%)				Sum
	Classification of Cell Diameter (μm)				
	I (5–100)	II (100–200)	III (200–300)	IV (300–400)	
300	22.8	18.6	15.6	22.5	79.5
600	27.5	21.1	17.4	23.8	89.8

**Table 2 polymers-11-00221-t002:** Experimental data of fully expanded thickness (units of mm) [14].

	${\dot{q}}_{\mathrm{irr}}^{\prime \prime }$ (kW/m^2^)	35	50	65
DFT (mm)				
2	Upper	22.3 (17.7)	22.4 (17.6)	22.9 (17.1)
	Lower	19.2 (20.8)	20.2 (19.8)	20.5 (19.5)
3	Upper	30.3 (24.7)	31.7 (23.3)	31.2 (23.8)
	Lower	29.0 (26.0)	29.7 (25.3)	31.0 (24.0)
4	Upper	-	41.2 (18.8)	-
	Lower	-	39.6 (20.4)	-

The number in the parentheses indicates the z-coordinate in Figure 6.

**Table 3 polymers-11-00221-t003:** View factors [38] and the corresponding true radiant heat absorptions (${\dot{q}}_{\mathrm{net}\_\mathrm{in}}^{\prime \prime }$) at emissivity (*ε*) = 0.77.

*z* (mm)	*F*_h-top_ *	${\dot{q}}_{\mathrm{net}\_\mathrm{in}}^{\prime \prime }$ (kW/m^2^)		
		${\dot{q}}_{\mathrm{irr}}^{\prime \prime }$ = 35 kW/m^2^	50 kW/m^2^	65 kW/m^2^
30	0.2382	24.75	35.36	45.97
25	0.2253	23.42	33.45	43.49
20	0.2126	22.10	31.57	41.04
15	0.2002	20.81	29.73	38.65

* For 100 mm^2^ specimens.

**Table 4 polymers-11-00221-t004:** Thermo-physical properties of the materials and convective coefficients in cone calorimeter testing.

	Properties	Conductivity *k* (W/(mK))	Density *ρ_d_* (kg/m^3^)	Specific Heat *c* (J/(kgK))	Emissivity *ε*	Convective Coefficient *h* (W/(m^2^ K))
Materials						
Intumescent coating		1.56	2077	1780	0.77 ^4^	14.6 ^5^
Steel plate ^1^		53.30	7870	440	-	
Air ^2^		0.03	1.16	1007	-	
Insulation board ^3^		0.21	900	1000	0.90	

^1^ Mild steel element (AISI C1020); ^2^ Standard air properties; ^3^ High-performance insulation board (Promatech-T produced by Promat, Tisselt, Belgium); ^4^ Hemispherical total emissivity [14]; ^5^ Coefficient for turbulent free convection adjacent to a horizontal upward-facing surface [14].

**Table 5 polymers-11-00221-t005:** Geometric variables used in the supplementary FEA simulations.

Classification	Φ ^1^	*L*^1^ (μm)	*N* _PV_ ^1^	*δ*_unit_^2^ (μm)	*δ*_cell_^2^ (μm)	*δ*_wall_^2^ (μm)
I	0.913	3000	60	50	48	1
II			20	150	143	3
III			12	250	239	6
IV			8	375	358	8

^1^ Independent variable; ^2^ Dependent variable.

**Table 6 polymers-11-00221-t006:** Results of supplementary simulations at Φ = 0.913 and ${\dot{q}}_{\mathrm{irr}}^{\prime \prime }$ = 50 kW/m^2^.

Heat Transfer Course	Effective Thermal Conductivity, *k*_eff_ (W/(mK))			
	Class I	Class II	Class III	Class IV
A-A’	0.1080	0.1105	0.1130	0.1160
B-B’	0.1061	0.1050	0.1040	0.1027
C-C’	0.0937	0.0932	0.0927	0.0922
D-D’	0.0953	0.0979	0.1005	0.1037

**Table 7 polymers-11-00221-t007:** *k*_eff_* for inorganic intumescent coating at Φ = 0.913 and ${\dot{q}}_{\mathrm{irr}}^{\prime \prime }$ = 50 kW/m^2^.

Temperature (°C)	*k*_eff_* (W/(mK))				Upper Bound	Lower Bound
	Heat Transfer Course					
	A-A’	B-B’	C-C’	D-D’		
300	0.1118	0.1045	0.0930	0.0993	0.1118	0.0930
600	0.1117	0.1045	0.0930	0.0991		

**Table 8 polymers-11-00221-t008:** Geometric variables for multicellular modelling.

*L*^1^ (μm)	Φ ^1^	*N* _PV_ ^1^	*δ*_unit_^2^ (μm)	*δ*_cell_^2^ (μm)	*δ*_wall_^2^ (μm)
3000	0.865	1	3000.0	2790.2	104.9
		2	1500.0	1395.1	52.5
		3	1000.0	930.1	35.0
		6	500.0	465.0	17.5
		11	272.7	253.7	9.5
		50	60.0	55.8	2.1
	0.895	1	3000.0	2838.1	80.9
		2	1500.0	1419.1	40.5
		3	1000.0	946.0	27.0
		6	500.0	473.0	13.5
		11	272.7	258.0	7.4
		50	60.0	56.8	1.6
	0.930	1	3000.0	2893.1	53.5
		2	1500.0	1446.5	26.7
		3	1000.0	964.4	17.8
		6	500.0	482.2	8.9
		11	272.7	263.0	4.9
		50	60.0	57.9	1.1

^1^ Independent variable; ^2^ Dependent variable.

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
