# Peer review of "Mechanism of Heat Transfer through Porous Media of Inorganic Intumescent Coating in Cone Calorimeter Testing"

_polymers, 2019, doi:10.3390/polym11020221_

Round 1

Reviewer 1 Report

This paper describes heat transfer process through a particular form of porous media: ‘inorganic’-based intumescent coating in full-expansion state. To quantitatively analyse its thermal insulation performance, a series of numerical simulations are primarily conducted using a Finite Element Analysis (FEA) software. To support FEA simulations and their verification, several tests on characteristics of the polymeric material are secondarily carried out utilising bench-scale instruments. However, before this manuscript is accepted for publication, it needs a minor revision for clarity and quality. The revision may include the following points:

The authors only listed two of their own publications in Introduction. Please improve the section adding more information about the polymer applications.

Regarding the porous media, what is the advantages of the polymer? 

The modeling theory is too long in this paper. Some information could be provided in supplementary information.

How could you verify the accuracy of the developed model with experimental data?

Please give the optimal parameters of the study.

Conclusions need to be reduced. What is the future possibility of this model?

Author Response

Please, find out an attached file which contains the authors’ response to the reviewer’s comments.

Reviewer 2 Report

The authors have contributed an appreciable work. This work is very important academically, industrially and in terms of safety point of view. Not much work was done to the extent that these authors have presented ultimately the work is accepted for publication. 

However, the authors should go through the manuscript again for minor English revisions and spell check. 

Author Response

(The authors gave the same response as above.)

Reviewer 3 Report

This study reports a comprehensive study on the porous intumescent coatings, including modeling and experimental work. Both studies are well written and validated with reasonable approachs. 

My only question about the procedure of determining the porosity of the intumescent during the CC tests are unclear. Those data are time/temperature dependent parameters which are hard to be measured during the tests. Those procedures need to be clarified in the revision.

Author Response

(The authors gave the same response as above.)
